# The Effects of Diabetes and Being Overweight on Patients with Post-COVID-19 Syndrome

**Simona Kirbiš [1], Nina Sobotkiewicz [1], Barbara Antolinc Schaubach [1], Jernej Završnik [2,3], Peter Kokol [4,5,6], Matej Završnik [7] and Helena Blažun Vošner [6,8,\*]**

1   Pneumophtisiology Department, General Health Organizational Unit, Community Healthcare Center dr. Adolf Drolc Maribor, 2000 Maribor, Slovenia; simona.kirbis@zd-mb.si (S.K.); nina.sobotkiewicz@zd-mb.si (N.S.); barbara.antolinc@zd-mb.si (B.A.S.)
2   Paediatric Department, Children and Youth Protection Unit, Community Healthcare Center dr. Adolf Drolc Maribor, 2000 Maribor, Slovenia; jernej.zavrsnik@zd-mb.si
3   Alma Mater Europaea—ECM, 2000 Maribor, Slovenia
4   Faculty of Electrical Engineering and Computer Science, University of Maribor, 2000 Maribor, Slovenia; peter.kokol@um.si
5   Faculty of Medicine, University of Maribor, 2000 Maribor, Slovenia
6   Scientific Research Department, Community Healthcare Center dr. Adolf Drolc Maribor, 2000 Maribor, Slovenia
7   Department of Endocrinology and Diabetology, Internal Medicine Clinic, University Clinical Center, 2000 Maribor, Slovenia; matej.zavrsnik@ukc-mb.si
8   Faculty of Health and Social Sciences, 2000 Slovenj Gradec, Slovenia
\*   Correspondence: helena.blazun@zd-mb.si

**Abstract:** In the aftermath of the COVID-19 pandemic, post-COVID-19 syndrome (PCS) remains a challenge and may continue to pose a major health problem in the future. Moreover, the influences of type 2 diabetes and being overweight on PCS remain unclear. This study aimed to assess these influences. We performed an observational study from October 2020 to July 2022, which included 466 patients (269 males and 197 females) with a median age of 65. They were hospitalized due to COVID-19 pneumonia and had persistent symptoms after 1 month of COVID-19 infection. The patients were divided into four groups according to the study objectives: patients with type 2 diabetes, overweight patients, overweight patients with type 2 diabetes, and average-weight patients without type 2 diabetes. The clinical and demographic data collected during hospitalization and regular visits to the Community Healthcare Center dr. Adolf Drolc Maribor were analyzed. Our results showed that type 2 diabetes patients had more difficult courses of treatment and longer hospitalizations. Moreover, more type 2 diabetes patients underwent rehabilitation than the other study groups. The prevailing symptoms of our patients with PCS were dyspnea and fatigue, mostly among female patients with type 2 diabetes. Our study also showed that more women with type 2 diabetes and overweight women with type 2 diabetes suffered from secondary infections. Furthermore, more overweight patients were treated in the intensive care unit than patients from the other groups. However, our study showed an interesting result: patients with type 2 diabetes had the shortest PCS durations. Type 2 diabetes and being overweight are risk factors for PCS onset and prolonged duration. Therefore, our data that revealed a shorter duration of PCS in type 2 diabetes patients than the other investigated groups was unexpected. We believe that answering the questions arising from our unexpected results will improve PCS treatment in general.

**Keywords:** post-COVID-19 syndrome; diabetes; overweight; COVID-19 pneumonia; observational study

## 1. Introduction

In the aftermath of the COVID-19 pandemic, post-COVID-19 syndrome (PCS) or 'long' COVID-19 remains a challenge and may continue to pose a major health problem in the future. The symptoms range from chronic fatigue and dyspnea to cognitive problems.

Obesity, diabetes, and being overweight have been global health problems and are key risk factors for various infections, including COVID-19. Zhou et al. reviewed the relationship between obesity, diabetes, and COVID-19 infection. They concluded that despite all the facts regarding altered immunity, the aggravation of inflammatory storms, and abnormalities in the lung physiology of diabetes patients, we still lack evidence that diabetes increases susceptibility to COVID-19 infection [1]. Obesity and being overweight are associated with altered pulmonary mechanics and physiology and increased angiotensin-converting enzyme 2 (ACE2) expression, which affects the acute phase of infection and progression to respiratory failure [1]. Diabetes and obesity alter pulmonary and lung functions. Diabetes predominantly affects the central and peripheral airway functions and the dissipative and elastic properties of respiratory tissues [2]. The most consistently reported effect of obesity on lung function is a reduction in functional residual capacity (FRC). Moreover, spirometric variables, such as forced expiratory volume in 1 s (FEV1) and forced vital capacity (FVC), tend to decrease with increasing body mass indices (BMIs) [3–5] However, this effect is small; the FEV1 and FVC are usually within the normal range in healthy obese adults [4,5].

Several studies have revealed that overweight patients and those with diabetes mellitus have an increased risk of developing PCS [6–9] PCS can be classified into two subtypes: post-acute COVID-19 with persistent symptoms lasting beyond four weeks after infection (PACS) and 'long' COVID-19 with irreversible tissue damage lasting beyond 12 weeks [8].

The influence of type 2 diabetes mellitus on PCS via various pathophysiological mechanisms has also been confirmed by Raveendran and Misra [10]. Lacavalerie et al. [11] found that over half of obese patients with PCS had a significant alteration in aerobic exercise capacity and a marked reduction in oxygen pulse. Moreover, these patients were more prone to pathological pulmonary limitations and pulmonary gas exchange impairment [12]. Picone et al. [13] showed that nutrition is closely linked to PCS due to inter-systemic homeostasis involving the endocrine, neurological, and immune–metabolic systems, particularly in obese and diabetic patients. Similar results, especially among patients older than 65, have been reported by Paneni and Patrono [14] and Shang et al. [15].

The purpose of our study was to determine whether similar PCS patterns also occur in Slovenia, especially in its northern region. This study included patients who were hospitalized at the University Clinical Centre of Maribor and then followed up in the Community Healthcare Center dr. Adolf Drolc. We focused on patients with diabetes and overweight patients and assessed how they differed from the reference population of patients without those conditions. We aimed to examine the differences and anomalies, not explain them. However, we want to encourage such research and plan to carry it out in future studies.

## 2. Materials and Methods

### 2.1. Study Population and Research Design

This retrospective observational study was conducted at the Community Healthcare Center dr. Adolf Drolc Maribor (HCM), Slovenia, from October 2020 to July 2022. All patients had previously been infected with SARS-CoV-2 and developed COVID-19 pneumonia that required hospitalization. They were regularly checked by physicians at the HCM within 3–4 weeks after discharge from the hospital and then every 3 months until post-COVID-19 symptoms disappeared. All 466 patients were retained in the study for the whole study period.

All patients with persistent symptoms after 1 month were selected as the study population for our analysis. Clinical and demographic data collected during hospitalization and the first HCM visit were analyzed. Physicians reviewed and collected data, including patient demographics, comorbidities, home medications, vital signs at hospital admission, duration of hospitalization, maximum oxygen requirement at hospitalization, treatment with remdesivir and dexamethasone (doses and protocols used were standard for all patients), radiological findings, admission to the intensive care unit (ICU), duration of ICU hospitalization, mechanical or non-invasive ventilation, bacterial superinfection, and vac-

cination status after vaccination was available. At regular checks after discharge from the hospital, the physicians performed a clinical evaluation, administered a questionnaire about post-COVID-19 symptoms (dyspnea, cough, chest pain, and fatigue), examined chest X-ray results, and conducted lung function tests.

Before conducting this study, we obtained the research participants' consent and carefully instructed them about the goals and purposes of our research. We also explained to them that participation was voluntary and that they could withdraw from the research at any time without providing a reason. Additionally, we clarified that only anonymized data would be used and these would be carefully protected by the principal researcher. The researchers of this study also obtained consent from the HCM Ethics Commission.

### 2.2. Statistical Analysis

Based on the study objectives, the patient population was divided into the following four groups:

- Patients who neither had diabetes nor were overweight (None group (NG));
- Overweight patients (Overweight group (OWG));
- Patients with type 2 diabetes (Diabetes group (DMG));
- Overweight patients with type 2 diabetes (Diabetes + Overweight group (DM + OWG)).

Being overweight was determined using BMI. All patients with a BMI greater than $25 \text{ kg/m}^2$ were classified as overweight. Most characteristics associated with PCS consisted of dichotomous information; therefore, the exact chi-square test was applied with each group as a grouping variable. The 16 analyzed variables (discretized lung function, symptoms, complications, and treatment) were coded as 1 or 0, corresponding to the presence or absence of the patient's features. The eight continuous variables (demographics and clinical assessment) were analyzed using the Kruskal–Wallis test with each group as the variable. The duration of PCS was analyzed using the Kaplan–Meier survival analysis.

Statistical analyses were performed using SPSS Statistics for Windows (version 28.0; IBM Corp., Armonk, NY, USA). A *p*-value of less than 0.05 was deemed statistically significant.

### 3. Results

Our sample consisted of 466 patients with post-COVID-19 syndrome. Among them, there were 269 males and 197 females. The males were between 30 and 92 years old, with a median age of 65. The females were between 19 and 98 years old, with a median age of 67. There were 77 patients in the NG (41 males and 36 females), 274 in the OWG (144 males and 130 females), 17 (12 males and 5 females) in the DMG, and 98 (72 males and 26 females) in the DM + OWG.

The Kruskal–Wallis test showed that body mass index (BMI) ($p < 0.001$), intensive care unit (ICU) duration ($p = 0.031$), and age differed significantly among the four groups in the male population. However, FVC%, FEV1%, FEV1/FVC%, and the duration of hospitalization for PCS did not significantly differ. In the female population, BMI significantly differed between the groups ($p < 0.001$).

The differences between the groups are shown in Table 1. The longest median PCS durations were observed in the OWG for males (5 months) and the DM + OWG for females (6 months). The longest median hospitalizations were observed in the DMG for males (13 days) and the NG and DM + OWG for females. The shortest median hospitalizations were observed in the OWG for males (8 days) and the DMG for females (6 days). The median PCS duration was generally shorter for males than for females. The DM + OWG had the highest median BMI among males (BMI = 31.00) and females (BMI = 33.9). Moreover, the lowest median BMIs were observed in the NG for males (BMI = 23.40) and the DMG for females (BMI = 22.8). The longest ICU median durations were observed in the DM + OWG, the OWG, and the NG for males (6 days) and the NG for females (11 days). The shortest ICU median durations were observed in the DMG for both males (0 days) and females (4 days). In general, the median ICU stay was longer for females than males.

**Table 1.** Median values for continuous PVC variables.

| Group | Gender | Statistics | HOS (Days) | ICU (Days) | FVC (%) | FEV 1 (%) | FEV1/FVC (%) | PCS (Months) | Age (Years) | BMI (kg/m²) |
|-------|--------|-----------|-----------|-----------|---------|-----------|-------------|-------------|-------------|-------------|
| None (NG) | Male (n = 41) | Median | 8.5 | 6 | 91 | 91 | 88 | 4 | 62 | 23.4 |
| | | Minimum | 2 | 1 | 48 | 25 | 50 | 1 | 30 | 18.3 |
| | | Maximum | 26 | 13 | 138 | 148 | 115 | 12 | 92 | 25.0 |
| | Female (n = 36) | Median | 11 | 11 | 95.5 | 94 | 96 | 5 | 66 | 23.0 |
| | | Minimum | 3 | 1 | 67 | 11 | 74 | 1 | 19 | 19.4 |
| | | Maximum | 41 | 21 | 131 | 131 | 112 | 12 | 98 | 25.0 |
| Overweight (OWG) | Male (n = 144) | Median | 8 | 6 | 88 | 90.5 | 93 | 5 | 63 | 30.5 |
| | | Minimum | 1 | 1 | 47 | 40 | 4 | 1 | 33 | 25.2 |
| | | Maximum | 57 | 27 | 134 | 135 | 127 | 12 | 88 | 44.3 |
| | Female (n = 130) | Median | 9 | 6 | 94 | 93 | 88 | 5 | 66 | 31.8 |
| | | Minimum | 2 | 2 | 54 | 31 | 4 | 1 | 30 | 25.2 |
| | | Maximum | 52 | 38 | 131 | 140 | 113 | 12 | 90 | 47.5 |
| Diabetes (DMG) | Male (n = 12) | Median | 13 | 0 | 88.5 | 87.5 | 83.5 | 3 | 74.5 | 24.1 |
| | | Minimum | 7 | 0 | 70 | 72 | 76 | 1 | 62 | 21.1 |
| | | Maximum | 32 | 0 | 103 | 116 | 106 | 8 | 90 | 24.8 |
| | Female (n = 5) | Median | 6 | 4 | 90 | 96 | 78 | 4 | 71 | 22.8 |
| | | Minimum | 4 | 4 | 78 | 79 | 77 | 3 | 60 | 20.2 |
| | | Maximum | 8 | 4 | 98 | 98 | 106 | 8 | 80 | 24.0 |
| Diabetes + Overweight (DM + OWG) | Male (n = 72) | Median | 11 | 6 | 85.5 | 88.5 | 82 | 4 | 69 | 31.0 |
| | | Minimum | 2 | 1 | 52 | 37 | 53 | 1 | 47 | 25.2 |
| | | Maximum | 95 | 49 | 118 | 118 | 112 | 12 | 87 | 47.8 |
| | Female (n = 26) | Median | 11 | 8 | 93 | 94 | 79 | 6 | 76 | 33.9 |
| | | Minimum | 2 | 3 | 70 | 67 | 65 | 1 | 48 | 25.3 |
| | | Maximum | 25 | 14 | 128 | 124 | 111 | 12 | 90 | 48.9 |

HOS—hospitalization, ICU—intensive care unit, FVC—forced vital capacity, FEV—forced expiratory volume in the first second, PCS—post-COVID-19 symptoms, BMI—body mass index.

The lowest FVC% values were observed in the DM + OWG for males (85.5%) and the DMG for females (90%). Overall, FVC% values were lower for males than females. The lowest median FEV1% values were observed in the DMG for males (87.5%) and the OWG for females (93%). The lowest median FEV1/FVC% values were observed in the DM + OWG for males (82%) and the DMG for females (78%). In general, female patients were older than male patients. Furthermore, the chi-square test for categorical variables revealed significant frequency differences between the four groups defined above among females only, namely in FEV1 < 85% ($p$ = 0.049), dexamethasone (DEXA) ($p$ = 0.012), and secondary bacterial infection ($p$ = 0.019).

Table 2 presents the percentages of patients with abnormal lung function, in-hospital treatment, complications during hospitalization, and PCS. Table 2 also reveals that the significant difference in FEV1 < 85% among females was due to the lower number of female patients in the DMG (FEV1% was 60.0%; there were five female patients) than other groups. The significant difference in DEXA was because only 52.1% of females were treated with DEXA in the NG. The most common secondary bacterial infections were found in the DMG (33.3%) and DM + OWG (28.6%) for females and the NG (15.4%) and OWG (12.3%) for males. The chi-square test of the categorical variables only revealed significant frequency differences between males and females in FVC < 85% ($p$ = 0.049), with more females having an FVC% lower than 85% than males. Additionally, significantly more females claimed fatigue ($p$ = 0.038) than males did.

**Table 2.** Frequencies for the categorical variables.

| Gender | Group | Lung Function | | Treatment | | | | | | | Complications | | | Post-COVID Symptoms | | | |
|---|---|---|---|---|---|---|---|---|---|---|---|---|---|---|---|---|---|
| | | FVC < 85% | FEV < 85% | DEXA | REM | ICS | ANT | REH | ICU | EMB | THR | MYP | BI | DYS | CP | Cough | FTG |
| Male | None | 61.0% | 58.5% | 71.2% | 39.2% | 33.3% | 38.0% | 3.9% | 9.8% | 3.8% | 0.0% | 25.0% | 15.4% | 46.2% | 5.8% | 34.6% | 23.1% |
| | Overweight | 63.2% | 68.1% | 73.9% | 33.1% | 29.7% | 25.8% | 6.8% | 16.7% | 8.0% | 1.8% | 23.3% | 12.3% | 50.3% | 16.0% | 43.6% | 31.3% |
| | Diabetes | 58.3% | 66.7% | 84.6% | 15.4% | 7.1% | 23.1% | 13.3% | 0.0% | 0.0% | 0.0% | 13.3% | 6.7% | 53.3% | 13.3% | 33.3% | 20.0% |
| | Diabetes + Overweight | 54.2% | 58.3% | 79.5% | 38.5% | 30.1% | 30.4% | 6.2% | 19.4% | 6.1% | 1.2% | 17.1% | 8.5% | 47.6% | 12.2% | 41.5% | 24.4% |
| | Total | 60.2% | 63.7% | 75.3% | 34.8% | 29.4% | 28.9% | 6.5% | 15.6% | 6.4% | 1.3% | 21.5% | 11.5% | 49.0% | 13.1% | 41.0% | 27.6% |
| Female | None | 80.6% | 77.8% | 52.1% | 28.9% | 18.8% | 34.0% | 10.2% | 5.6% | 10.2 | 0.0% | 24.5% | 24.5% | 59.2% | 20.4% | 42.9% | 36.7% |
| | Overweight | 78.5% | 76.9% | 71.4% | 34.7% | 27.3% | 21.8% | 6.3% | 13.1% | 6.8% | 0.0% | 24.3% | 11.5% | 53.4% | 16.9% | 40.5% | 37.8% |
| | Diabetes | 60.0% | 60.0% | 87.5% | 28.6% | 11.1% | 28.6% | 11.1% | 20.0% | 0.0% | 0.0% | 22.2% | 33.3% | 55.6% | 33.3% | 55.6% | 33.3% |
| | Diabetes + Overweight | 73.1% | 61.5% | 82.4% | 44.1% | 32.4% | 39.4% | 6.5% | 15.4% | 5.7% | 0.0% | 22.9% | 28.6% | 42.9% | 20.0% | 34.3% | 40.0% |
| | Total | 77.7% | 73.2% | 69.6% | 34.8% | 25.6% | 26.9% | 7.3% | 12.2% | 7.1% | 0.0% | 24.1% | 17.4% | 53.1% | 18.7% | 40.7% | 37.8% |

FVC—forced vital capacity, FEV—forced expiratory volume in the first second, DEXA—dexamethasone, REM—remdesivir, ICS—inhalation corticosteroid, REH—rehabilitation, ICU—intensive care unit, EMB—embolia, THR—thromboses, MYP—myopathy, BI—bacterial infection, DYS—dyspnea, CP—chest pain, FTG—fatigue.

Overall, our study showed that within the treatment variables, most patients were treated with DEXA, and the fewest patients underwent rehabilitation. Males were more likely to be treated with all forms of therapy except rehabilitation. The group most frequently treated with DEXA was the DMG (87.5%) for females and the DMG (84.6%) for males. Furthermore, most female patients treated with remdesivir (REMDE) were in the DM + OWG (44.1%), and males were in the NG (39.2%). Inhalation corticosteroid (ICS) and antibiotic (ANT) treatments were mostly used for males in the NG (33.3% (ICS), 38.0% (ANT)) and for females in the DM + OWG (32.4% (ICS), 39.4% (ANT)). In all groups, the percentage of patients who underwent rehabilitation was very low. The highest frequencies of both sexes in the DMG were 13.3% for males and 11.1% for females.

Generally, hospital complications occur more frequently among females than among males. Among males, we found that more myopathies (NG (25%), OWG (23.3%)) were observed than other complications during hospitalization. Among complications for females, bacterial superinfections (DMG (33.3%), DM + OWG (28.6%)) prevailed.

The most prevalent PCS symptoms were dyspnea and rare chest pain. Dyspnea was most prevalent among the females of the NG (59.2%); however, the groups did not differ significantly. Moreover, chest pain (33.3%) and coughing (55.6%) among the females of the DMG and fatigue among the females of the DM + OWG (40%) were the highest. Among males, the most prevalent symptom was dyspnea (49%), and the highest occurrence was in the DMG (53.3%).

Our study showed that most patients treated in the ICU were in the DMG for females (20%) and the DM + OWG for males (19.4%). However, the lowest number of patients were treated in the ICU (9.8% males and 5.6% females) in the NG. Additionally, we found that more men had a lower FVC% in the DM + OWG than in the other groups.

Summarizing the median values of our two target groups (DMG and DM + OWG), we can conclude that males in the DMG had higher hospitalization values and FEV1% than females. On the other hand, males and females in the DM + OWG had the highest BMIs, and females had the longest PCS durations. Males in the DMG had the lowest ICU and PCS durations and FVC%. Women in the DMG had the lowest ICU, hospitalization, and PCS durations, as well as FVC%, FEV1/FVC%, and BMIs.

Regarding frequencies, males in the DMG were the prevalent group for DEXA treatment, undertaking rehabilitation, and experiencing dyspnea. The females in the DMG were the prevalent group for DEXA treatment, undertaking rehabilitation, bacterial infections, chest pain, coughing, and experiencing fatigue. The least prevalent group of males included REMDE treatment, time spent in the ICU, antibiotic consumption, and complications such as embolia, thrombosis, myopathy, bacterial infection, and PCS fatigue. The least prevalent group of females also included REMDE treatment, time spent in the ICU, and additional complications such as embolia, thrombosis, myopathy, and PCS fatigue. On the other hand, the results for males in the DM + OWG did not stand out, except that it was the least prevalent group regarding undertaking rehabilitation. However, females in the DM + OWG were the most prevalent group for REMDE treatment, time spent in the ICU, consuming antibiotics, and fatigue as PCS symptoms. This group was also the least prevalent for complication thromboses and PCS symptoms, such as dyspnea, chest pain, and coughing.

The Kaplan–Meier analyses shown in Figures 1 and 2 reveal the surprising fact that both males and females with diabetes had the shortest PCS durations; in other words, all patients recovered within 8 months. Other groups had longer PCS durations, meaning that after the 12th month, some patients still did not recover, namely 13.1% in the Obesity group, 12.5% in the Diabetes + Obesity group, and 4.6% in the None group. The mean recovery times were 5.3 ± 3.2 in the None group, 5.7 ± 3.1 in the Obesity group, 3.9 ± 2.2 in the Diabetes group, and 5.28 ± 3.3 in the Diabetes + Obesity group.

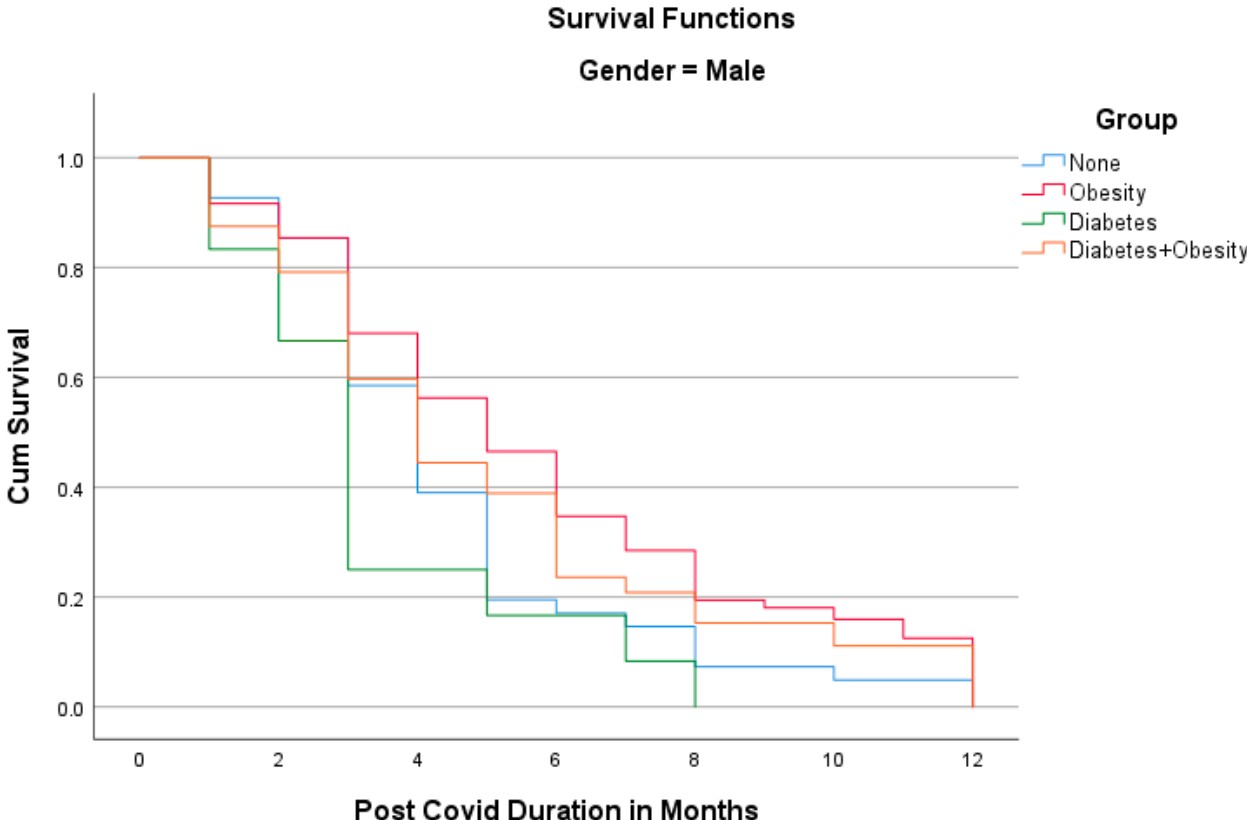

**Figure 1.** The Kaplan–Meier analysis of post-COVID syndrome duration for males.

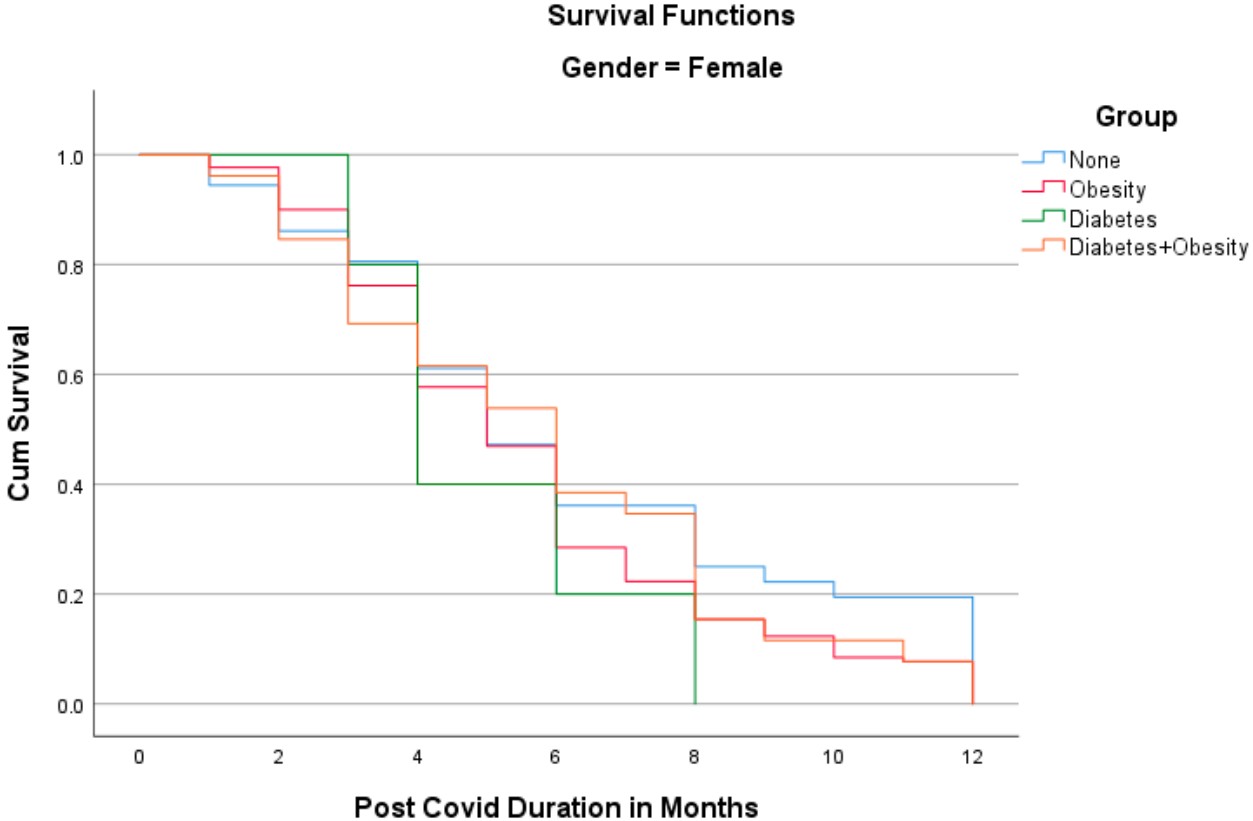

**Figure 2.** The Kaplan–Meier analysis of post-COVID syndrome duration for females.

## 4. Discussion

In our study, PCS was shorter among patients who only had type 2 diabetes (DMG) than the other three groups. The type 2 diabetes group (DMG), both men and women, had the shortest durations of PCS at 3 and 4 months, respectively, and which was significantly shorter than overweight males and females with diabetes, who had the longest durations of PCS at 5 and 6 months, respectively. The shorter duration of PCS among patients with type 2 diabetes than among the other investigated groups in our study was unexpected.

While most researchers mention diabetes and obesity as risk factors for PCS onset or its prolonged duration, some researchers disagree. Su et al. [16] associated pre-existing type 2 diabetes with PCS in their study. Multiple early factors predict post-acute COVID-19 sequelae. Patients with diabetes have significantly more fatigue as part of PCS. Moreover, diabetes and metabolic diseases increase the risk of developing PCS [17]. Additionally, a small study from India showed that patients with type 2 diabetes have significantly higher levels of fatigue than those with pre-diabetes [18].

In contrast, a Spanish study that analyzed coughing in PCS 1 year after COVID-19 pneumonia did not find any association with diabetes. The authors concluded that the metabolic impact of diabetes was more important in the acute phase of COVID-19 [19]. Moreover, a meta-analysis by Sudhakar et al. [20] collected 11 studies that linked diabetes and post-acute COVID-19 for at least 4 weeks and 14 studies that did not. Therefore, there is no clear answer to whether diabetes is an important risk factor for PCS.

The average length of treatment in the ICU in our study was 6 days for DMG females, but DMG males were not admitted to the ICU. This contrasts with the general finding in our study that ICU treatment was longer for men. The lack of treatment in the ICU for DMG males was surprising and could indicate the possibility of survival bias, which we cannot refute with the current data. Secondary bacterial infections were common among DMG females. There were practically no complications (myocarditis, embolisms, and thrombosis) in the type 2 diabetes group for both sexes, despite complications occurring in the other groups.

The lowest average FVC% value was found among DMG females. This also contrasts with the higher average FVC% values among females. A high median FEV 1% value was observed among DMG females, but the lowest was observed among DMG males, which agrees with the average higher general FEV 1% value for females. Additionally, the lowest FEV1/FVC% value was observed among DMG females. However, it is difficult to explain these observations. The DMG was small, with 5 females and 12 males. The age of the females in this group was the lowest on average (66 years), but the age of the males in the type 2 diabetes group was the second highest (74.5 years). Women with type 2 diabetes were the youngest and spent the longest time in the ICU, which may indicate a higher morbidity of women in our group.

In addition, women with type 2 diabetes (DMG females) had some of the lowest FVC% and FEV1/FVC% values, with higher FEV1% values as part of lung-function monitoring. Men in the type 2 diabetes group (DMG males) did not significantly differ in lung function parameters.

Based on analyzing data in the field of treatment, our study showed that type 2 diabetes patients had fewer chronic lung diseases, such as asthma and chronic obstructive pulmonary disease (COPD), compared to the other groups. Therefore, they also needed fewer days of treatment in the ICU. We also found that type 2 diabetes patients had more difficult courses of treatment and longer hospitalizations; therefore, they needed more rehabilitation time than the other groups. Moreover, we found that more patients with type 2 diabetes were treated with dexamethasone than other groups, and the difference was especially evident among female patients. According to these results, type 2 diabetes could have more influence on the acute phase of COVID-19 disease and its early complications, including pneumonia, and less influence on PCS. However, more intensive treatment and monitoring by physicians for type 2 diabetes patients during hospitalization and

rehabilitation after COVID-19 pneumonia may influence the duration of PCS and shorten it compared to other groups of patients.

A possible limitation of our study is the small number of patients in the type 2 diabetes group compared to the other three groups. However, the group was statistically large enough, and non-parametric tests were used, which are less sensitive to the sample size. The second possible limitation is that we did not consider the influence of morbidity. However, considering general statistical distributions, it is very unlikely that our DMG was bipolar, i.e., having patients who recovered fully from PCS on one extreme and having patients who died in the acute phase on the other. Furthermore, our study dealt with PCS; therefore, patients who died before the detection of PCS would not have contributed to the knowledge about its long course. Moreover, our study was observational, and analyzing causalities was beyond its scope [21]. The third possible limitation is that HbA1c was not evaluated. However, this study revealed some interesting phenomena that require further investigation.

Additionally, this study was relatively small for collecting data on symptoms, as we only collected data for dyspnea, coughing, chest pain, and fatigue. These four symptoms are crucial in PCS treatment and are on the list provided by Ballering et al. [22].

The advantage of our research is that it is based on clinical practice in a real-world healthcare-system environment, which is uniform for practically all citizens of Slovenia; therefore, the follow-up of patients in our center was systematic. Therefore, generalizing the data for the entire country is possible because Slovenia is a very small country and, therefore, quite uniform in most health determinants. The data were collected by a small but qualified and dedicated group of medical professionals from one center; therefore, we considered the data to be of high quality regarding their accuracy.

The outcomes of our study are comparable to similar studies performed worldwide, which might be because COVID-19 management in Slovenia did not differ from other countries. However, we found that PCS duration was shorter among patients who only had type 2 diabetes than the other three groups. Recent research showed that the onset of long COVID could be associated with metabolic dysfunction [23–25]. A study performed in the same region reports that metformin is often the first line of treatment for diabetes patients [26]. Metformin has been shown to have a potential protective role in COVID-19 [27] and long COVID [28,29] so it might be the contributing factor for the shorter duration of PCS among diabetic patients. We currently do not know how to explain the above finding conclusively. Nevertheless, we would like to draw attention to it and encourage studies that would more precisely define and perhaps elucidate this observed fact. Thus, further research will be dedicated to the influence of type 2 diabetes treatment and rehabilitation on the duration of PCS and other pathological mechanisms that might affect the duration of PCS.

**Author Contributions:** S.K., H.B.V. and P.K. designed the research methodology; S.K., N.S. and B.A.S. performed the study and collected data; S.K., H.B.V., P.K., J.Z. and M.Z. prepared the manuscript; P.K. performed the statistical analysis. All authors have read and agreed to the published version of the manuscript.

**Funding:** This research received no external funding.

**Institutional Review Board Statement:** This study was conducted in accordance with the Declaration of Helsinki and approved by the Institutional Ethics Committee of Community Healthcare Center dr. Adolf Drolc Maribor (Zdravstveni dom dr. Adolfa Drolca Maribor), 02/010/03-002/01/22. (approved on 16 February 2022).

**Informed Consent Statement:** Informed consent was obtained from all subjects involved in this study.

**Data Availability Statement:** The data presented in this study are available upon request from the corresponding author. The data are not publicly available due to data privacy restrictions.

**Conflicts of Interest:** The authors declare that the research was conducted in the absence of any commercial or financial relationships that could be construed as potential conflict of interest.

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
