# Peer review of "The Effects of Diabetes and Being Overweight on Patients with Post-COVID-19 Syndrome"

_2036-7449, doi:10.3390/idr15060067_

Round 1

Reviewer 1 Report (Previous Reviewer 3)

Comments and Suggestions for Authors

The manuscript looks good. It can accepted for publication.

Author Response

Thank you for your comments

Reviewer 2 Report (Previous Reviewer 1)

Comments and Suggestions for Authors

The authors have covered all my observations. Congratulations

Author Response

Thankyou to enable us to make the paper better

This manuscript is a resubmission of an earlier submission. The following is a list of the peer review reports and author responses from that submission.

Round 1

Reviewer 1 Report

Comments and Suggestions for Authors

-Check cites style

-Adjust Table 1 to the journal's requirements and add the p-value of each analysis. Maybe you have to readjust the whole table for a more fluid presentation.

-Table 2 would be better in graphs for easier understanding.

-Figures 1 and 2 would be better in a single graph comparing all the variables between sexes, even adding some statistical analysis contrasting that people with diabetes have a shorter duration of PCS.

-I am surprised that people with diabetes have a shorter duration of PCS. Do you know if your patients have controlled DM2? Or out of control? What treatment did you use to control DM2?

Some drugs used to control DM2 (metformin) have been seen to reduce the risk of mortality and the severity of COVID-19. In addition, it has been seen that in other RNA viruses (Dengue and ZIKV), it helps to reduce the viral load. A similar mechanism has also been proposed for the case of SARS-CoV-2. If you know these data, it would enormously enrich your article.

Farfan-Morales, C.N., Cordero-Rivera, C.D., Osuna-Ramos, J.F. et al. The antiviral effect of metformin on zika and dengue virus infection. Sci Rep 11, 8743 (2021). https://doi.org/10.1038/s41598-021-87707-9

Outpatient treatment of COVID-19 and incidence of post-COVID-19 condition over 10 months (COVID-OUT): a multicentre, randomised, quadruple-blind, parallel-group, phase 3 trial. Bramante, Carolyn TAnderson, Blake et al. The Lancet Infectious Diseases, Volume 0, Issue 0

Therapeutic Potential of Metformin in COVID-19: Reasoning for Its Protective Role. Samuel, Samson Mathews et al. Trends in Microbiology, Volume 29, Issue 10, 894 - 907

Comments on the Quality of English Language

Minor editing of English language required

Author Response

Thank you for your valuable comments, we adapted the paper accordingly

-Check cites style

References have been checked and corrected

-Adjust Table 1 to the journal's requirements and add the p-value of each analysis. Maybe you have to readjust the whole table for a more fluid presentation.

Only descriptive analysis has been used; thus, no p-values could be computed. According to the author's guidelines, tables will be adjusted by the publisher

-Table 2 would be better in graphs for easier understanding.

Some values are very small and would not be seen well on the graph

-Figures 1 and 2 would be better in a single graph comparing all the variables between sexes, even adding some statistical analysis contrasting that people with diabetes have a shorter duration of PCS.

The male and female Kaplan Meier survival curves almost overlap, so curves would not be distinguishable. The statistics has been aded

-I am surprised that people with diabetes have a shorter duration of PCS. Do you know if your patients have controlled DM2? Or out of control? What treatment did you use to control DM2?

While our study was observational, and we didn’t had the DM2 treatment in our data collection protocol we don’t have the concrete data about the application of metformin in targeted population. However in another study which some of the authors conducted we encountered that currently metformin is the main drug used in DM2. Consequently, we mentioned the possible effect of metformin on the PCS duration in the discusion

Reviewer 2 Report

Comments and Suggestions for Authors

Dear Authors

I have carefully read the scientific article, which has much scientific merit and solidity, contributing to the existing body of knowledge on this subject. Only I recommend that the limitation that glycocyte hemoglobin (HbA1c) was not evaluated should be explicitly highlighted.

Author Response

Thank you for your comment, we extended the limititations with your sugestion

Reviewer 3 Report

Comments and Suggestions for Authors

The authors have made an observation that post covid symptoms are less severe in patients with diabetes, perhaps reported for the first time, and in a new population, which is interesting

However, the results are presented in confusing manner and there is no possible explanation for these differences that they observe, which are contrary to published results. 

It would be worthwhile to understand or have reasonable basis to explain for these observations.

Too many variables and too many confusing terminology are making it difficult to fully comprehend the author's views. 

Please review the writing to enable better understanding and clarity of thoughts. 

Comments on the Quality of English Language

Please review the writing to enable better understanding and clarity of thoughts. 

Author Response

I would like to thank the reviewer for the useful comments

The authors have made an observation that post covid symptoms are less severe in patients with diabetes, perhaps reported for the first time, and in a new population, which is interesting

Thank you

However, the results are presented in confusing manner and there is no possible explanation for these differences that they observe, which are contrary to published results.  It would be worthwhile to understand or have reasonable basis to explain for these observations.

The possible explanation was added to the new version of the paper

Too many variables and too many confusing terminology are making it difficult to fully comprehend the author's views. 

The paper was rewritten and restructured and was edited by a professional editing service

Please review the writing to enable better understanding and clarity of thoughts.

The paper was rewritten and restructured and was edited by a professional editing service